# Distribution of Cerebrovascular Phenotypes According to Variants of the *ENG* and *ACVRL1* Genes in Subjects with Hereditary Hemorrhagic Telangiectasia

**DOI:** 10.3390/jcm11102685

**Published:** 2022-05-10

**Authors:** Eleonora Gaetani, Elisabetta Peppucci, Fabiana Agostini, Luigi Di Martino, Emanuela Lucci Cordisco, Carmelo L. Sturiale, Alfredo Puca, Angelo Porfidia, Andrea Alexandre, Alessandro Pedicelli, Roberto Pola

**Affiliations:** 1HHT Center, Fondazione Policlinico Universitario A. Gemelli IRCCS, Università Cattolica del Sacro Cuore, 00168 Rome, Italy; elisabetta.peppucci@gmail.com (E.P.); fabiana.agostini@libero.it (F.A.); luigidimartino7@gmail.com (L.D.M.); emanuela.luccicordisco@policlinicogemelli.it (E.L.C.); carmelo.sturiale@policlinicogemelli.it (C.L.S.); alfredo.puca@policlinicogemelli.it (A.P.); angelo.porfidia@policlinicogemelli.it (A.P.); andrea.alexandre@policlinicogemelli.it (A.A.); alessandro.pedicelli@policlinicogemelli.it (A.P.); roberto.pola@unicatt.it (R.P.); 2Department of Medicine, Fondazione Policlinico Universitario A. Gemelli IRCCS, Università Cattolica del Sacro Cuore, 00168 Rome, Italy; 3Department of Neurosurgery, Fondazione Policlinico Universitario A. Gemelli IRCCS, Università Cattolica del Sacro Cuore, 00168 Rome, Italy; 4Department of Genetics, Fondazione Policlinico Universitario A. Gemelli IRCCS, Università Cattolica del Sacro Cuore, 00168 Rome, Italy; 5Department of Radiology, Fondazione Policlinico Universitario A. Gemelli IRCCS, Università Cattolica del Sacro Cuore, 00168 Rome, Italy

**Keywords:** hereditary hemorrhagic telangiectasia, cerebrovascular malformations, arteriovenous malformations, genetics, gene variants, *ENG*, *ACVRL1*

## Abstract

Hereditary Hemorrhagic Telangiectasia (HHT) is an autosomal dominant disorder caused, in more than 80% of cases, by mutations of either the endoglin (*ENG*) or the activin A receptor-like type 1 (*ACVRL1*) gene. Several hundred variants have been identified in these HHT-causing genes, including deletions, missense and nonsense mutations, splice defects, duplications, and insertions. In this study, we have analyzed retrospectively collected images of magnetic resonance angiographies (MRA) of the brain of HHT patients, followed at the HHT Center of our University Hospital, and looked for the distribution of cerebrovascular phenotypes according to specific gene variants. We found that cerebrovascular malformations were heterogeneous among HHT patients, with phenotypes that ranged from classical arteriovenous malformations (AVM) to intracranial aneurysms (IA), developmental venous anomalies (DVA), and cavernous angiomas (CA). There was also wide heterogeneity among the variants of the *ENG* and *ACVRL1* genes, which included known pathogenic variants, variants of unknown significance, variants pending classification, and variants which had not been previously reported. The percentage of patients with cerebrovascular malformations was significantly higher among subjects with *ENG* variants than *ACVRL1* variants (25.0% vs. 13.1%, *p* < 0.05). The prevalence of neurovascular anomalies was different among subjects with different gene variants, with an incidence that ranged from 3.3% among subjects with the c.1231C > T, c.200G > A, or c.1120C > T missense mutations of the *ACVRL1* gene, to 75.0% among subjects with the c.1435C > T missense mutation of the *ACVRL1* gene. Further studies and larger sample sizes are required to confirm these findings.

## 1. Introduction

Hereditary hemorrhagic telangiectasia (HHT) is a rare autosomal dominant genetic disorder with an estimated prevalence of 1–5 cases/10,000 individuals [1]. The clinical hallmarks of the disease are multiple arteriovenous shunts, or arteriovenous malformations (AVMs), affecting several organs, including the skin, nasal and oral mucosa, lungs, liver, brain, and gastrointestinal tract [2,3].

Most HHT-causing variants are detected in endoglin (*ENG*) and activin A receptor type II-like kinase 1 (*ALK1/ACVRL1*) genes [4,5,6]. Less than 2% of cases are instead due to variants of the Family Member 4 (*SMAD4*) gene [7]. Pathogenic variants in the *ENG* and *ACVRL1* genes result in different HHT types, commonly referred to as HHT1 and HHT2, respectively [4,5] Pathogenic variants in the *SMAD4* gene give rise to a combined syndrome of HHT and juvenile polyposis (JP): the JP-HHT syndrome [7]. Several hundred variants of the *ENG* and *ACVRL1* genes, including deletions, duplications, missense and nonsense mutations, splice defects, and silent variants, have been identified in patients with HHT [8,9] and are present in HHT mutation databases (http://arup.utah.edu/database/ENG/ENG_welcome.php, accessed on 7 March 2022) [10]. Whether single gene variants are associated with specific HHT phenotypes is under investigation and remains to be determined, since HHT is a rare disease and many of the above-mentioned gene variants have been reported in the literature only once.

At the level of the central nervous system (CNS), there is a wide spectrum of possible cerebrovascular malformations in HHT. These include AVMs, but also cavernous malformations (CAs), capillary telangiectasias (CTs), developmental venous anomalies (DVAs), arteriovenous fistulas (AVFs), and intracranial aneurysms (IAs) [11,12].

The aim of this study was to assess the distribution of different types of cerebrovascular malformations according to specific variants of the *ENG* and *ACVRL1* genes in subjects with HHT.

## 2. Materials and Methods

*Ethics.* This study was approved by the Ethics Committee of the Fondazione Policlinico Universitario A. Gemelli IRCCS (protocol number 6241/20, ID 2999, date of approval 20 February 2020). Due to the retrospective nature of the study, and the fact that the study only consisted of analysis of data available in the electronic database of our hospital, additional informed consent was not required.

*Patients.* We collected data of patients followed at the HHT Center of the Fondazione Policlinico Universitario A. Gemelli IRCCS of Rome, Italy. Only data of patients who had a genetically established diagnosis of HHT and had undergone Magnetic Resonance Imaging of the brain with angiographic study (MRA) were collected.

*Review of MRA.* MRA images were reviewed by four independent investigators of our HHT Center (two neuroradiologists and two neurosurgeons), blinded to the genotypes of patients. The results of the review were tabulated as follows: negative exam (no evidence of cerebrovascular malformations); positive exams (evidence of at least one of the following cerebrovascular malformations: AVM, CA, CT, DVA, AVF, IA).

*Statistical analysis.* Results are presented as mean ± SD or number and percentage. Comparisons between groups were made by using Student’s t test. Differences were considered statistically significant for *p* < 0.05.

## 3. Results

We identified 77 patients with genetically verified HHT with fully accessible MRA images of the brain, which is part of the screening routinely performed on HHT patients at our center. The demographic, clinical, genetic, and neuroradiological characteristics of these patients are reported in Table 1. Briefly, all patients were Italians of Caucasian ancestry. Mean age was 53.6 ± 16.7 years. Male/female ratio was 33/44. Most patients (98.7%) belonged to a definite HHT family, while only one patient (1.3%) had negative family history. Regarding HHT clinical hallmarks, recurrent epistaxis and cutaneous telangiectasias were present in 94.8% and 90.9% of patients, respectively. Lung and liver involvement was found in 32.5% and 44.1% of patients, respectively. GI bleeding was reported in 24.7% of cases. *ENG* and *ACVRL1* variants were found in 16 (20.8%) and 61 (79.2%) patients, respectively. MRAs were negative for any type of cerebrovascular malformations in 65 patients (84.4%), while they were positive in 12 patients (15.6%), with a male/female ratio of 5/7. There were four patients (three males and one female) with an AVM, three patients (one male and two females) with an IA, one patient (male) with the concomitant presence of an AVM and an IA, two patients (2 females) with a CA, and two patients (two females) with a DVA, for a total of 13 vascular malformations. 

A schematic representation of the diverse types of cerebrovascular malformations found in our cohort is shown in Figure 1.

Figure 2 presents instead representative MR images of the diverse types of cerebrovascular malformations found in our HHT cohort. 

Table 2 reports the different gene variants that we found in our cohort. In total, there were 10 *ENG* variants and 24 *ACVRL1* variants. Among the *ENG* variants, seven were known pathogenic variants (which were found in a total of 13 patients), one was a variant pending classification, and two were variants not previously reported in the HHT mutation database. Among the *ACVRL1* variants, 11 were known pathogenic variants (distributed in a total of 45 patients), three were variants pending classification (distributed in a total of three patients), one was a variant of unknown significance (VUS), and eight were variants not previously reported in the HHT mutation database (distributed in a total of 11 patients). The most represented gene variants in our population were the c.1231C > T (14 patients from five unrelated families), the c.200G > A (eight patients from three unrelated families), and the c.1120C > T (eight patients from three unrelated families) missense mutations of the *ACVRL1* gene.

Table 3 reports the distribution of cerebrovascular malformations according to *ENG* and *ACVRL1* variants. The percentage of patients with cerebrovascular malformations was significantly higher among subjects with *ENG* variants (four out of 16 patients) than among subjects with *ACVRL1* variants (eight out of 61 patients) (25.0% vs. 13.1%, respectively, *p* < 0.05). Among the four patients with *ENG* variants who had cerebrovascular malformations, three were females and one was a male. Among the eight patients with *ACVRL1* variants who had cerebrovascular malformations, four were females and four were males. When the correlation analysis was limited to known pathogenic variants, the percentage of patients with cerebrovascular malformations was 23.1% in the *ENG* group and 11.1% in the *ACVRL1* group. As mentioned above, the c.1231C > T, the c.200G > A, and the c.1120C > T missense mutations of the *ACVRL1* gene were the most represented variants in our population. Taken together, they were present in 30 patients, i.e., 38.9% of the study population. However, in these 30 patients we found only one cerebrovascular anomaly, with an incidence of 3.3%. On the opposite side, there was the c.1435C > T missense mutation of the *ACVRL1* gene that, although present only in four patients, was associated with three cases of cerebrovascular malformations, with an incidence of 75.0%. There was also the c.771dup variant of the *ENG* gene that, although found only in three patients, was associated with one case of cerebrovascular malformations (incidence 33.3%).

## 4. Discussion

In this study, we evaluated the distribution of cerebrovascular phenotypes according to specific *ENG* and *ACVRL1* gene variants in subjects with HHT. It is interesting to note that, in our population, neurovascular anomalies were significantly more frequent among *ENG* than *ACVRL1* patients. This is consistent with the results of a recent meta-analysis [13], showing that patients with HHT1 have a significantly higher brain AVM prevalence compared with those with HHT2 and confirms that different genetic backgrounds may have an impact on the cerebrovascular phenotypes of HHT patients.

On the other hand, it is also notable that some gene variants, although numerically well represented in our population, were never–or very rarely–associated with the presence of a cerebrovascular malformation. In particular, this was the case of the c.1231C > T, c.200G > A, and c.1120C > T missense mutations of the *ACVRL1* gene, which cumulatively were present in 24 patients (35.3% of our population) but were associated with a cerebrovascular malformation only in one case. It is also interesting that, among HHT2 patients, a consistent proportion of cerebrovascular malformations (37.5%) was found in patients carrying gene variants for which an established pathogenic role does not exist yet, according to the HHT Mutation Database [10]. Indeed, in this study, we found nine novel variants which had never been reported previously. One of these was a deletion (c.582_600del) detected in the *ENG* gene in one patient who did not have any cerebrovascular malformation at MRA of the brain. The other eight variants which had never been reported before were in the *ACVRL1* gene and were found in a total of 10 patients. Additional studies will be important for establishing whether the relationship between these variants and cerebrovascular malformations are common or coincidental occurrences.

This study also constitutes a useful report on the incidence of cerebrovascular malformations in HHT, which was 15.6% in our population. This incidence is similar to that reported by Woodall and coll. in 2014 (18.6%) and Brinjikji and coll. in 2016 [11,14]. It is instead higher than that reported by Maher and coll. in 2001 (3.7%) [15], but it should be noted that this study took into account patients who were not screened for the presence of a cerebral vascular malformation, while our study only included subjects for whom neurovascular imaging was available. Regarding the type of malformation, our study confirms that vascular brain lesions are heterogeneous among HHT patients. AVMs were the most common in our population, with an incidence of 6.5%. IAs were also common, with an incidence of 5.2%. The notion that HHT patients often have aneurysmatic diseases has been recently confirmed by a retrospective analysis of Ring and coll., who found 43 patients with at least one aneurysm on 418 HHT subjects, with a prevalence of 10.3% [16]. Taken together, IAs and AVMs accounted for 75.0% of all cerebrovascular malformations. The remaining malformations were CAs (two cases, with an incidence of 2.6% in the study population), and DVAs (one case, with an incidence of 1.3% in the study population).

This study has potential limitations. First, the sample size is small, and many gene variants are presented only in single patients. This certainly hinders the possibility to evaluate whether an association exists between these variants and specific cerebrovascular malformations. Additionally, it is known that the rupture of cerebrovascular malformations is devastating in the HHT pediatric population [17,18]. Since our study only includes adult individuals, it might be limited by a survival bias. Additionally, we have been able to analyze only those patients with genetically confirmed HHT who had undergone MRA. Therefore, our sample is not representative of the general HHT population. In addition, since the gold standard for the diagnosis of arteriovenous shunts is catheter angiography and we only had access to MRA data, the incidence of AVFs and microscopic AVMs might be underestimated.

In summary, this study provides updated information on the incidence and distribution of cerebrovascular malformations according to specific gene variants in subjects with HHT. Further studies are needed to substantiate these findings and potentially lead to personalization of risk stratification and screening regimens for cerebrovascular malformations in HHT patients.

## Figures and Tables

**Figure 1 jcm-11-02685-f001:**
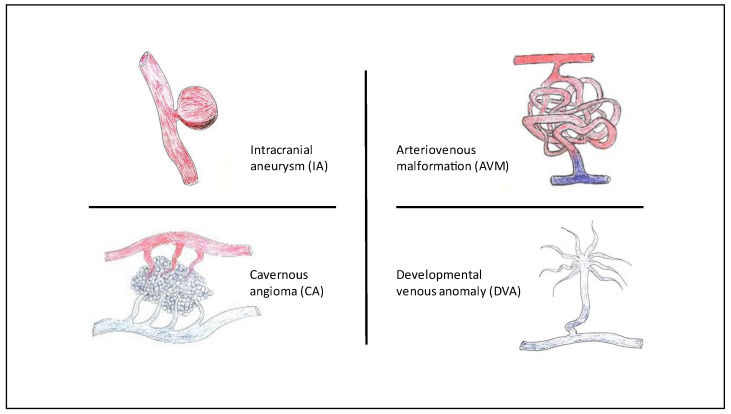
Types of cerebrovascular malformations found in our cohort of HHT patients.

**Figure 2 jcm-11-02685-f002:**
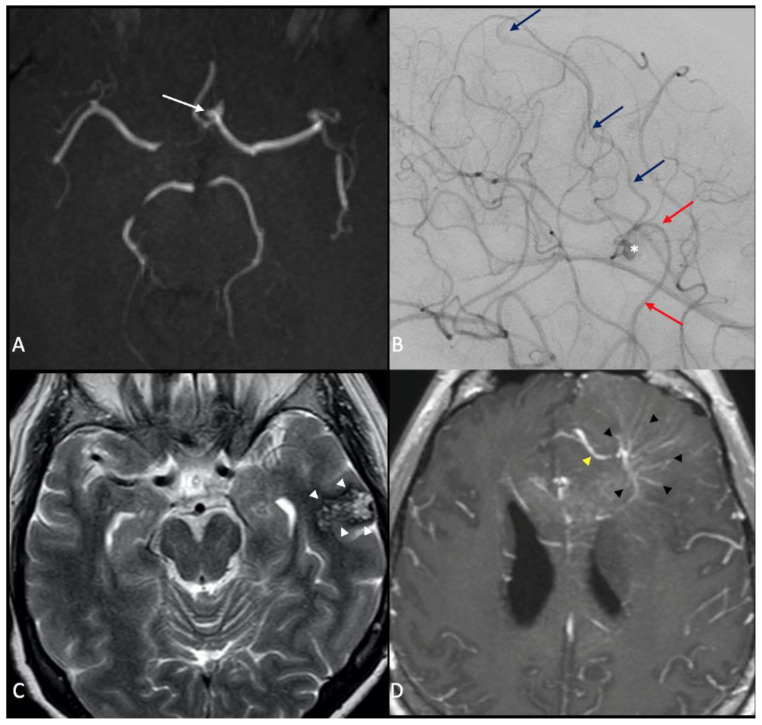
Representative MR images of the cerebrovascular malformations found in our cohort. (**A**) 3D-time of flight MR demonstrating a small anterior communicating artery aneurysm. (**B**) Lateral projection in a Digital Subtraction Angiography during internal carotid artery injection: a small nidus of an arteriovenous malformation (AVM) is evident (white asterisk), as well as an arterial feeder coming from the middle cerebral artery (red arrows) and the draining vein going to the superior sagittal sinus (blue arrows). (**C**) MR T2-weighted axial image shows a large, cavernous angioma (CA) in the cortical region of the left temporal lobe (white arrow heads), with a characteristic “popcorn” appearance and a rim of signal loss due to hemosiderin. (**D**) MR T1-weghted post-contrast axial image showing a vascular malformation characterized by a “caput medusae” (black arrowheads), which is a collection of dilated medullary veins, which converges in an enlarged transcortical or collector vein (yellow arrowhead), which is pathognomonic of developmental venous anomaly (DVA).

**Table 1 jcm-11-02685-t001:** Characteristics of the study population.

Mean age (years±SD)	53.6 ± 16.7
Gender (male/female ratio)	33/44
Epistaxis (n/total)	73/77
Mucocutaneous telangiectases (n/total)	70/77
Family history of HHT (n/total)	76/77
Pulmonary AVMs (n/total)	25/77
Hepatic AVMs (n/total)	34/77
Gastrointestinal AVMs (n/total)	19/77
Cerebrovascular malformations (n/total) - Arteriovenous malformation (AVM) - Intracranial aneurysm (IA) - Developmental venous anomaly (DVA) - Cavernous angioma (CA) - AVM + IA	12/77 4/12 3/12 2/12 2/12 1/12
*ENG* mutations	16/77
*ACVRL1* mutations	61/77

**Table 2 jcm-11-02685-t002:** Distribution of gene variants.

*ENG*			
Nucleotide Change	Type of Variant	Classification	Patients (n)
c.771dup	Duplication	pathogenic	4
c.309_311del	Deletion	pathogenic	3
c.1678C > T	Nonsense	pathogenic	2
c.1199delG	Deletion	pathogenic	1
c.1 A > G	Missense	pathogenic	1
c.511C > T	Nonsense	pathogenic	1
c.967_968del	Deletion	pathogenic	1
c.1088G > A	Missense	pending classification	1
c.582_600del	deletion	not previously reported	1
c.1115T > C	Missense	not previously reported	1
** *ACVRL1* **			
**Nucleotide Change**	**Type of Variant**	**Classification**	**Patients (n)**
c.1231C > T	Missense	pathogenic	14
c.1120C > T	Missense	pathogenic	8
c.200G > A	Missense	pathogenic	8
c.1435C > T	Missense	pathogenic	4
c.430C > T	Missense	pathogenic	4
c.1121G > A	Missense	pathogenic	1
c.1135G > A	Missense	pathogenic	2
c.1232G > A	Missense	pathogenic	1
c.218_219insAA	Insertion	pathogenic	1
c.435del	Deletion	pathogenic	1
c.1435C > T	Missense	pathogenic	1
c.230G > A	Missense	pending classification	1
c.526G > T	Missense	pending classification	1
c.988G > T	Missense	pending classification	1
c.1445C > A	Missense	VUS	2
c.264C > G	Missense	not previously reported	2
c.847_853delinsTT	deletion-insertion	not previously reported	2
c.144_145insG	Insertion	not previously reported	1
c.771_772dup	Duplication	not previously reported	1
c.837C > G	Missense	not previously reported	1
c.1028A > C	Missense	not previously reported	1
c.398C > G	Missense	not previously reported	1
c.915_916dup	Duplication	not previously reported	1
c.738_739dup	Duplication	not previously reported	1

**Table 3 jcm-11-02685-t003:** Distribution of cerebrovascular malformations according to *ENG* and *ACVRL1* variants.

Nucleotide Change	Positive/Total MRA	Type of Malformation
*All ENG variants*	4/16 (25.0%)	
*Pathogenic variants*	3/13 (23.1%)	
c.1199delG	0/1 (0.0%)	--
c.1678C > T	1/2 (50.0%)	AVM
c.1A > G	1/1 (100.0%)	IA
c.309_311del	0/3 (0.0%)	--
c.511C > T	0/1 (0.0%)	--
c.771dup	1/4 (25.0%)	AVM
c.967_968del	0/1 (0.0%)	--
*Other variants*	1/3 (33.3%)	
c.1115T > C	1/1 (100.0%)	DVA
c.1088G > A	0/1 (0.0%)	--
c.582_600 del	0/1 (0.0%)	--
*All ACVRL1 variants*	8/61 (13.1%)	
*Pathogenic variants*	5/45 (11.1%)	
c.1120C > T	0/8 (0.0%)	--
c.1121G > A	0/1 (0.0%)	--
c.1135G > A	0/2 (0.0%)	--
c.1231C > T	1/14 (7.1%)	CA
c.1232G > A	1/1 (100.0%)	AVM + IA
c.1435C > T	3/4 (75.0%)	2 IA (M, F), 1 DVA
c.200G > A	0/8 (0.0%)	--
c.218_219insAA	0/1 (0.0%)	--
c.430C > T	0/4 (0.0%)	--
c.435del	0/1 (0.0%)	--
*Other variants*	3/16 (18.7%)	
c.230G > A	0/1 (0.0%)	--
c.526G > T	1/1 (100.0%)	AVM
c.988G > T	0/1 (0.0%)	--
c.1445C > A	0/2 (0.0%)	--
c.144_145insG	0/1 (0.0%)	--
c.771_772dup	1/1 (100.0%)	CA
c.837C > G	0/1 (0.0%)	--
c.1028A > C	0/1 (0.0%)	--
c.264C > G	0/2 (0.0%)	--
c.398C > G	0/1 (0.0%)	--
c.915_916dup	1/1 (100.0%)	AVM
c.847_853delinsTT	0/2 (0.0%)	--
c.738_739dup	0/1 (0.0%)	--

## Data Availability

Data are available upon request to the corresponding author.

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
