# Peer review of "Distribution of Cerebrovascular Phenotypes According to Variants of the ENG and ACVRL1 Genes in Subjects with Hereditary Hemorrhagic Telangiectasia"

_jcm, 2022, doi:10.3390/jcm11102685_

Round 1

Reviewer 1 Report

The manuscript by Gaetani et al, is very useful in terms of the presence of brain arteriovenous malformations, in HHT patients and their putative correlation with the type of mutations. It is well presented and described, but some small changes and additions would improve the manuscript.

Major: A figure showing the different types of CAVM described by the authors is needed to illustrate the changes observed in the HHT population.

Other changes to include in the text:

  1. All the words which are representing genes, should be written in italics, i.e ENG, ACVRL1, SMAD4, all over the manuscript, please.
  2. In the abstract, there is a sentence which should be rephrased to: we have  analyzed retrospectively collected images of (page 1 line 16)
  3. Page 1 last line, 43: change to different HHT "types", instead of phenotypes. Currently we said 2 types of HHT, the phenotype is the clinical characteristics of each patient individually considered.
  4. Page 2 in line 47, after the references 8,9 and after the "," add: "including hundreds of deletions, duplications,  missense and nonsense mutations,  splice defects, and  silent variants, are present in HHT......", to make it easier the description of the changes.
  5. In Mat and Methods section, in line 65,  the suggestion is adding after hospital, an additional informed consent was not required. Since the patients probably had already a previous informed consent from the first time of screening (clinical, genetics)
  6. In line 70 in this same Mat and Methods section page 2, I think that you intended to have a subsection named Review of MRA. I advice to introduce this section separately.
  7. Results. page 2. This is a question whose answer, I would like to be included at the beginning of the results. Is MRA done as part of the HHT screening, once the genetics analysis of HHT has been confirmed?  Is it part of the screening routine of the clinically diagnosed patients as HHT? This is an important issue, specially in neonates, when they are genetically HHT, in an HHT family because of the devastating effects that CAVMs may have at this early age, as you comment later on in the discussion section.
  8. Page 3 line 105. It is necessary to clarify whether the most represented gene variants are found in 1 single family, 2 independent families, or in how many independent families. For example, in 104 line, it is written that the most represented gene variants in our population were the c.1231C>T (14 patients). How were these patients distributed. Were they belonging to the same family? Were they from 2 different non related families? This is important to add in each of the numbers you mention. How many independent families?
  9. Discussion page 6 line 139. instead of that has shown that, to avoid repetition of "that" you could  say: analysis, showing that patients.....
  10. In the same discussion page, 6, line 150 change as determined by interrogating the HHT., by " according to the HHT ....

This is all, I do congratulate authors by the study. This is well done.

Author Response

Response to Reviewer 1

The manuscript by Gaetani et al. is very useful in terms of the presence of brain arteriovenous malformations, in HHT patients and their putative correlation with the type of mutations. It is well presented and described, but some small changes and additions would improve the manuscript.

“Major: a figure showing the different types of CAVM described by the authors is needed to illustrate the changes observed in the HHT population.”

We thank the Reviewer for this suggestion. In the revised manuscript, we have included a figure (Figure 1) which presents the different type of cerebrovascular malformations found in our patient cohort.

1. “All the words which are representing genes, should be written in italics, i.e ENG, ACVRL1, SMAD4, all over the manuscript, please.”

This change has been made throughout the manuscript.

2. “In the abstract, there is a sentence which should be rephrased to: we have analyzed retrospectively collected images of (page 1 line 16)”

The sentence has been rephrased as requested.

3. “Page 1 last line, 43: change to different HHT "types", instead of phenotypes. Currently we said 2 types of HHT, the phenotype is the clinical characteristics of each patient individually considered.”

This change has been made.

4. “Page 2 in line 47, after the references 8,9 and after the "," add: "including hundreds of deletions, duplications,  missense and nonsense mutations,  splice defects, and  silent variants, are present in HHT......", to make it easier the description of the changes.

The sentence has been changed as suggested by the Reviewer.

5. “In Mat and Methods section, in line 65, the suggestion is adding after hospital, an additional informed consent was not required. Since the patients probably had already a previous informed consent from the first time of screening (clinical, genetics)

The suggested sentence has been added in the revised manuscript.

6. “In line 70 in this same Mat and Methods section page 2, I think that you intended to have a subsection named Review of MRA. I advice to introduce this section separately.”

A separate paragraph, named Review of MRA has been created in the revised manuscript. In this section, we also specify that there were 2 neuroradiologists and 2 neurosurgeons who reviewed the images.

7. “Results. page 2. This is a question whose answer, I would like to be included at the beginning of the results. Is MRA done as part of the HHT screening, once the genetics analysis of HHT has been confirmed? Is it part of the screening routine of the clinically diagnosed patients as HHT? This is an important issue, especially in neonates, when they are genetically HHT, in an HHT family because of the devastating effects that CAVMs may have at this early age, as you comment later on in the discussion section.”

In the revised version of the manuscript, we specify that MRA of the brain is part of the screening routinely performed on HHT patients in our center (page 2, lines 80-81).

8. Page 3 line 105. It is necessary to clarify whether the most represented gene variants are found in 1 single family, 2 independent families, or in how many independent families. For example, in 104 line, it is written that the most represented gene variants in our population were the c.1231C>T (14 patients). How were these patients distributed? Were they belonging to the same family? Were they from 2 different nonrelated families? This is important to add in each of the numbers you mention. How many independent families?

This is a very important point. In the revised version of the manuscript (page 4, lines 109-112), we specify that “the c.1231C>T was found in 14 patients from 5 unrelated families, the c.200G>A in 8 patients from 3 unrelated families, and the c.1120C>T in 8 patients from 3 unrelated families.

9. Discussion page 6 line 139. instead of that has shown that, to avoid repetition of "that" you could say: analysis, showing that patients.....

The suggested change has been made.

10. In the same discussion page, 6, line 150 change as determined by interrogating the HHT., by " according to the HHT ....

The suggested change has been made.

Reviewer 2 Report

Overall, I found this manuscript to be informative and succinct in its description of the correlation between ENG and ACVRL1 variants and cerebrovascular malformations. I believe this is an important extension of the available data and begins to give the field a better understanding of potential links between certain HHT variants and the various types of cerebrovascular malformations. The manuscript is well-written and the data is easily accessible. The authors do not use self-citations. I have no major concerns, only minor suggestions that might improve an already nice body of work.

Minor points:

1) In Table 1, under the Cerbrovascular malformations portion, “2/12” and “1/12” corresponding to Cavernous angioma (CA) and AVM + IA need to be aligned properly.

2) The data pertaining to the heterogeneity of cerebrovascular malformations and the association to the types of variants is very interesting. I wonder if it would be even more informative by adding the gender associated with each individual that displayed a cerebrovascular malformation. Perhaps this could be included in Table 3? The authors could then comment as to whether they did or didn’t see a sex difference correlation as well.

3) In the discussion (lines 142-155), the authors talk about how the most common variants (primarily ACVRL1) are not associated with malformations, while unknown pathogenic variants comprise a higher portion of patients with malformations. It might be useful to add an ending sentence, specifically in this paragraph, that states additional studies will be important for establishing whether the relationship between these variants and cerebrovascular malformations are common or coincidental occurrences.

4) I’m not as familiar with this aspect, but if there are other papers that refer to the different types of cerebrovascular malformations (IA, CA, etc.) associated with HHT, it might be helpful to add a few references and comment on how the authors’ findings compare to previous works. For example, is the prevalence of IA in HHT1 or HHT2, similar to other studies, etc.

Minor grammatical suggestions:

1) In the Introduction, change the second sentence (line 36-39) to the following (reduce the use of “the”):

The clinical hallmarks of the disease are multiple arteriovenous shunts, or arteriovenous malformations (AVMs), affecting several organs, including the skin, nasal and oral mucosa, lungs, liver, brain, and gastrointestinal tract [2-3].

2) In the Materials and Methods section, the 3rd sentence (line 70), which states “Review the MRA.”, should probably be taken out.

3) In the first paragraph of the Results section, the sentence beginning on line 89 could be slightly altered to the following:

MRAs were negative for any type of cerebrovascular malformation in 65 patients (84.4%), while they were positive in 12 patients (15.6%).

4) In the third paragraph of the Results section, the sentence beginning on line 124 could be slightly altered to the following by taking out “the”:

Taken together, they were present in 30 patients, i.e., 38.9% of the study population.

Author Response

Response to Reviewer 2

Overall, I found this manuscript to be informative and succinct in its description of the correlation between ENG and ACVRL1 variants and cerebrovascular malformations. I believe this is an important extension of the available data and begins to give the field a better understanding of potential links between certain HHT variants and the various types of cerebrovascular malformations. The manuscript is well written, and the data is easily accessible. The authors do not use self-citations. I have no major concerns, only minor suggestions that might improve an already nice body of work.

Minor points:

  1. “In Table 1, under the Cerbrovascular malformations portion, “2/12” and “1/12” corresponding to Cavernous angioma (CA) and AVM + IA need to be aligned properly.”

Done.

  1. “The data pertaining to the heterogeneity of cerebrovascular malformations and the association to the types of variants is very interesting. I wonder if it would be even more informative by adding the gender associated with each individual that displayed a cerebrovascular malformation. Perhaps this could be included in Table 3? The authors could then comment as to whether they did or didn’t see a sex difference correlation as well.”

We agree with the Reviewer that it would be interesting to analyze possible association between gender, gene variants, and cerebrovascular malformations. However, we think that the numbers are too small to be analyze in the setting of our study. To answer to the question raised by the Reviewer, in the revised manuscript, we have added information regarding the gender of the subjects affected by cerebrovascular malformations (from page 2, line 90 to page 3, line 94 and on page 6, lines 121-123), without making any comment on these data.

  1. “In the discussion (lines 142-155), the authors talk about how the most common variants (primarily ACVRL1) are not associated with malformations, while unknown pathogenic variants comprise a higher portion of patients with malformations. It might be useful to add an ending sentence, specifically in this paragraph, that states additional studies will be important for establishing whether the relationship between these variants and cerebrovascular malformations are common or coincidental occurrences.”

The suggested sentence has been included in the revised manuscript (page 8, lines 159-161).

  1. I’m not as familiar with this aspect, but if there are other papers that refer to the different types of cerebrovascular malformations (IA, CA, etc.) associated with HHT, it might be helpful to add a few references and comment on how the authors’ findings compare to previous works. For example, is the prevalence of IA in HHT1 or HHT2, similar to other studies, etc.

In the revised version of the manuscript, we have added two references (refs 14 and 16) that deal with the incidence of cerebrovascular malformations in HHT. We have also included additional comments on how our findings correlate with the literature.

Minor grammatical suggestions:

  1. In the Introduction, change the second sentence (line 36-39) to the following (reduce the use of “the”): The clinical hallmarks of the disease are multiple arteriovenous shunts, or arteriovenous malformations (AVMs), affecting several organs, including the skin, nasal and oral mucosa, lungs, liver, brain, and gastrointestinal tract [2-3].
  2. In the Materials and Methods section, the 3rd sentence (line 70), which states “Review the MRA.”, should probably be taken out.
  3. In the first paragraph of the Results section, the sentence beginning on line 89 could be slightly altered to the following: MRAs were negative for any type of cerebrovascular malformation in 65 patients (84.4%), while they were positive in 12 patients (15.6%).
  4. In the third paragraph of the Results section, the sentence beginning on line 124 could be slightly altered to the following by taking out “the”: Taken together, they were present in 30 patients, i.e., 38.9% of the study population.

The suggested changes have been made.

Round 2

Reviewer 1 Report

I liked all your changes, and I am very happy to see how you worked to improve the version, however, I would prefer in addition to the drawings  illustration a real image of MRI showing each type of lesion. This is the only thing you need to add to the article.

Author Response

“I liked all your changes, and I am very happy to see how you worked to improve the version, however, I would prefer in addition to the drawings illustration a real image of MRI showing each type of lesion. This is the only thing you need to add to the article.”

As requested, we have added a new figure (Fig. 2, page 4, lines 102-117) with representative MR images of the 4 types of cerebrovascular lesions found in our HHT cohort.